# Predictive Factors for Gait Recovery in Patients Undergoing Total Hip Arthroplasty: A Propensity Score Weighting Analysis

**DOI:** 10.3390/jcm14061979

**Published:** 2025-03-14

**Authors:** Yuna Kim, Seo Young Kim, Sung Ryul Shim, Jung Keun Hyun

**Affiliations:** 1Department of Rehabilitation Medicine, College of Medicine, Dankook University, Cheonan 31116, Republic of Korea; kimyuna727@dkuh.co.kr (Y.K.); juliet8383@naver.com (S.Y.K.); 2Department of Biomedical Informatics, College of Medicine, Konyang University, Daejeon 35365, Republic of Korea; 3Konyang Medical Data Research Group-KYMERA, Konyang University Hospital, Daejeon 35365, Republic of Korea; 4Department of Nanobiomedical Science and BK21 NBM Global Research Center for Regenerative Medicine, Dankook University, Cheonan 31116, Republic of Korea; 5Institute of Tissue Regeneration Engineering, Dankook University, Cheonan 31116, Republic of Korea

**Keywords:** total hip arthroplasty, gait recovery, rehabilitation protocols, propensity score weighting, postoperative muscle strength

## Abstract

**Objectives:** This paper’s objective was to identify clinical predictors, especially modifiable ones, associated with postoperative gait recovery in total hip arthroplasty (THA) patients, utilizing propensity score weighting (PSW) to control confounding factors. **Methods:** This retrospective cohort study included 221 patients who underwent primary unilateral THA. We used PSW analysis to balance patient characteristics. Univariate and multivariate logistic regression analyses were applied to determine predictors of improved gait recovery, assessing variables such as age, gender, and postoperative muscle strength. **Results:** Independent predictors of favorable gait recovery were male gender (Odds Ratio [OR]: 1.382; 95% Confidence Interval [CI]: 1.225–1.560; *p* < 0.001), younger age (OR: 0.990 per year; 95% CI: 0.985–0.995; *p* < 0.001), and postoperative hip flexor muscle strength greater than grade 3 (OR: 1.516; 95% CI: 1.177–1.953; *p* = 0.002). Muscle strength emerged as a modifiable factor, suggesting that targeted rehabilitation may enhance functional outcomes. **Conclusions:** Enhancing hip flexor strength postoperatively could significantly improve gait recovery in THA patients. These findings support developing individualized rehabilitation strategies to optimize functional outcomes.

## 1. Introduction

Total hip arthroplasty (THA) is the gold standard for managing severe hip joint diseases like osteoarthritis, rheumatoid arthritis, and certain hip fractures. This surgery alleviates pain, improves mobility, and restores function, enhancing patient quality of life [1]. Over recent decades, the prevalence of THA has steadily increased worldwide, largely driven by an aging population and advancements in surgical techniques and implant materials. Currently, more than 1 million THA procedures are performed annually on a global scale, with projections indicating continued growth. For instance, in the United States, approximately 512,000 THA procedures were performed in 2020, and this number is expected to rise to 850,000 by 2030 [2,3,4].

THA generally improves functional outcomes such as walking speed, stride length, and gait symmetry, but the extent of recovery varies considerably among individuals [5,6,7,8]. This variability complicates rehabilitation planning, making it essential to identify modifiable factors that influence gait recovery and can be targeted for intervention. Mobility, broadly defined as a patient’s ability to move independently in daily life, encompasses both gait performance and functional capacity in activities such as stair climbing and transfers. While objective gait parameters (e.g., stride length, walking speed) provide detailed biomechanical insights, functional gait classifications are more relevant in clinical decision-making [9].

Several factors influence postoperative gait recovery, including patient-specific characteristics such as age, severity of hip osteoarthritis, and preoperative mobility status [10,11]. Other factors, including comorbid conditions, perioperative management strategies, and muscle strength, may also contribute to recovery trajectories [8,12].

Previous studies have identified age, gender, and preoperative functional status as key determinants of mobility outcomes, but these factors are not modifiable, limiting their clinical utility [13,14,15]. Conversely, muscle strength—particularly hip flexor strength—is a modifiable factor that may play a pivotal role in gait recovery. Hip flexor weakness has been linked to shorter step length, reduced walking speed, and prolonged double-limb support time, all of which impair functional mobility [16,17]. However, the extent to which rehabilitation strategies can optimize these modifiable factors remains unclear, necessitating further investigation [18,19].

In addition, THA patients differ in their underlying hip pathology, which may influence the gait recovery trajectory. Patients undergoing THA for fracture-related conditions often experience acute functional decline, limited preoperative rehabilitation, and a different postoperative recovery course than patients undergoing elective THA for osteoarthritis or avascular necrosis [20,21]. Despite these differences, few studies have directly compared how fracture and non-fracture THA patients differ in their postoperative gait performance. This study addresses this gap by analyzing these groups separately.

Given the limitations of traditional statistical methods in observational studies, propensity score weighting (PSW) is used to adjust for confounders and to balance patient characteristics between groups. While cluster analysis has been explored in previous research to classify recovery patterns, PSW is better suited to reduce bias while maintaining direct group comparisons [22].

Thus, the aim of this study is to identify clinical predictors, particularly modifiable ones, associated with postoperative gait recovery after THA. By using PSW to control for confounding bias, this study provides a more accurate estimate of the independent effects of these predictors [23]. Understanding these factors may facilitate the development of personalized rehabilitation strategies to improve mobility and optimize long-term recovery outcomes in THA patients.

## 2. Materials and Methods

### 2.1. Study Design and Setting

This study is a retrospective cohort analysis designed to identify modifiable clinical factors associated with postoperative gait recovery following THA. The PSW method was employed to control for confounding variables inherent in observational studies, ensuring a balanced comparison between patient groups. This study adhered to the principles outlined in the 1964 Helsinki Declaration and its subsequent amendments.

### 2.2. Participants/Study Subjects

This study included 221 patients who underwent primary unilateral THA at Dankook University Hospital from January 2020 to December 2022. Given the retrospective design, no priori sample size calculation was performed. However, post hoc power analysis indicates that with 221 patients, this study achieves over 80% power to detect moderate effect sizes (odds ratios ≥ 1.5) for key predictors in the logistic regression model, assuming a 5% type I error rate [24]. To examine the impact of different underlying conditions on postoperative gait recovery, patients were categorized based on the etiology of their hip pathology into two classifications: those with fracture-related conditions and those with non-fracture conditions. The fracture classification included patients who required THA following acute hip fractures, such as femoral neck or intertrochanteric fractures resulting from trauma. In contrast, the non-fracture classification comprised patients who underwent THA due to chronic degenerative conditions, such as osteoarthritis, rheumatoid arthritis, or avascular necrosis. Classifying patients into fracture and non-fracture categories was crucial because the underlying cause of hip pathology significantly influences rehabilitation strategies and potential gait recovery. Patients with fracture-related conditions often experience different recovery trajectories due to the acute onset and severity of their injuries, while those with non-fracture conditions have distinct rehabilitation needs due to the chronic nature of their joint degeneration. This classification enabled a focused analysis of clinical predictors of gait recovery and helped to identify modifiable factors that could be optimized through specific rehabilitation interventions tailored to each condition. This study included adults who underwent unilateral THA due to fracture or non-fracture conditions and completed postoperative rehabilitation. Exclusion criteria included incomplete clinical data, a history of significant prior hip surgeries other than THA, pre-existing neurological disorders affecting gait, concomitant lower limb injuries, or the inability to complete rehabilitation due to severe comorbidities or cognitive impairments.

### 2.3. Description of Experiments, Treatment, or Surgery

The data collected in this study encompassed several key domains relevant to understanding postoperative gait recovery following THA. Demographic data, such as gender and age, were collected to identify potential baseline differences that might influence recovery outcomes. Postoperative rehabilitation included structured physical therapy sessions designed to restore range of motion, strengthen hip and lower extremity muscles, and improve gait mechanics. Patients participated in progressive weight-bearing exercises, functional range of motion training, and stair climbing exercises. Therapy intensity and duration were adjusted based on individual recovery progress and clinical assessments.

### 2.4. Variables, Outcome Measures, Data Sources, and Bias

Medical history information, including the underlying diseases and the duration of preoperative gait disturbances, was gathered to assess the impact of pre-existing conditions on postoperative rehabilitation outcomes. Treatment-related data, such as the duration of rehabilitation and the time interval from surgery to the initiation of rehabilitation, were also documented to evaluate their effects on recovery trajectories. Although the Physical Activity Scale for the Elderly (PASE) or other validated physical activity markers were not utilized due to the retrospective nature of this study, preoperative functional status and rehabilitation duration were included as surrogate measures of physical activity. These variables provide an indirect estimation of patients’ baseline mobility and level of rehabilitation participation, allowing for an assessment of their potential influence on gait recovery. Clinical assessments focused on evaluating postoperative muscle strength of the operated hip, specifically in flexion and abduction. This assessment was conducted using a standardized manual muscle testing method, graded on a scale from 0 to 5 [25], by physiatrists two days after surgery. A strength grade of 3 was chosen as a key threshold, as it indicates the ability to move against gravity, which is essential for fundamental weight-bearing activities such as standing and walking. Lower grades reflect insufficient strength for independent ambulation, while higher grades denote increased force generation that supports improved gait mechanics and recovery. This early assessment aimed to establish a baseline measure of muscle function that could be used to track recovery progress and identify patients who might benefit from targeted rehabilitation strategies. The clinical predictors selected for analysis, along with their rationale based on prior literature, are detailed in Appendix A.

This study focused on clinician-assessed gait ability rather than self-reported mobility measures, as the latter may not always align with actual performance in structured rehabilitation settings. Clinician assessments provide objective, standardized evaluations that more accurately reflect functional mobility in postoperative rehabilitation. Functional outcomes were primarily evaluated through assessments of gait ability, both preoperatively and postoperatively. Preoperative gait ability was assessed via structured interviews conducted by physiatrists in consultation with patients and their caregivers to provide a baseline measure of mobility and independence, lasting approximately 10–15 min. The structured interviews included specific questions regarding the patient’s ability to perform activities of daily living, such as stair climbing and ambulation at home without assistance. Postoperative gait ability was clinically evaluated by physiatrists at the conclusion of rehabilitation therapy, using direct observation and assessment of functional mobility. Based on their ability to perform daily activities, patients were classified into two groups: those with ‘good gait ability’ (G group) and those with ‘poor gait ability’ (P group). The G group included patients capable of performing essential daily activities independently, such as stair ascent and descent and independent home ambulation without assistance. The P group included those unable to achieve independent home ambulation, indicating a level of functional limitation requiring further rehabilitation support. These classifications align with established rehabilitation criteria used to assess functional independence and postoperative mobility recovery in THA patients [11,26].

### 2.5. Data Processing and Propensity Score Weighting (PSW)

To address selection bias, PSW was used to adjust for baseline differences between traumatic and non-traumatic groups. PSW was selected to retain the full sample size and balance covariates effectively, simulating the conditions of a randomized controlled trial [27]. This method allowed for a more accurate estimation of the association between clinical factors and postoperative gait recovery following THA.

Propensity scores were calculated using a logistic regression model, estimating the probability of each patient being classified as either fracture or non-fracture based on a set of covariates. These covariates included continuous variables such as age, gender, duration of preoperative gait disturbance, duration of rehabilitation, time interval from surgery to rehabilitation initiation, and postoperative muscle strength of the operated hip, as well as categorical variables, including preoperative gait ability and postoperative gait ability classification (G group or P group). The dependent variable in the propensity score model was the type of hip pathology (fracture vs. non-fracture). Propensity scores, ranging from 0 to 1, represented the likelihood of each patient being classified as either fracture or non-fracture given their baseline characteristics.

To evaluate the effectiveness of PSW in balancing the baseline characteristics between the fracture and non-fracture classifications, we compared covariates before and after weighting. The balance of covariates was assessed using standardized mean differences (SMD), with an SMD of less than 0.1 considered indicative of a good balance between classifications [28,29].

Data analyses related to PSW were performed using R version 4.3.1 (The R Foundation for Statistical Computing, Vienna, Austria), with the ‘survey’ package employed for PSW and covariate adjustment. For cases with missing data on key outcome variables, complete case analysis was utilized, excluding cases with incomplete data from the final analysis to ensure statistical consistency.

### 2.6. Statistical Analysis

Following the application of PSW, univariable and multivariable logistic regression analyses were conducted to identify clinical factors associated with good postoperative gait ability in THA patients. Univariable analyses utilized independent t-tests for continuous variables and chi-square tests or Fisher’s exact tests for categorical variables. Multivariable logistic regression was performed to adjust for potential confounders and determine independent predictors of good gait ability, defined by classification into the G group. The strength of associations was expressed as odds ratios (ORs) with 95% confidence intervals (CIs). All statistical analyses were two-tailed and a *p*-value < 0.05 was considered statistically significant.

## 3. Results

### 3.1. Characteristics of Participants Before and After PSW Prior to PSW

Significant differences in basic characteristics were observed between the fracture (*n* = 73) and non-fracture (*n* = 148) classifications. Participants in the fracture classification were significantly older, with a mean age of 68.33 ± 8.68 years compared to 61.73 ± 12.75 years in the non-fracture classification (*p* < 0.001). The fracture classification also had a higher proportion of patients capable of independent ambulation without an assistive device (86.3% vs. 72.3%) and fewer patients with muscle strength greater than grade 3 for hip flexion (23.3% vs. 39.9%, *p* = 0.015) and abduction (24.7% vs. 41.2%, *p* = 0.016), indicating notable baseline differences in key clinical characteristics. PSW effectively balanced baseline differences between the fracture and non-fracture groups. Following PSW adjustment, the weighted sample comprised 84 patients in the fracture group and 150.5 patients in the non-fracture group. The fractional value reflects the statistical weighting applied to individual participants rather than an actual patient count, a characteristic of PSW used to achieve covariate balance in observational studies. After weighing, no statistically significant differences in baseline characteristics remained (all *p* > 0.05), as shown in Table 1.

This outcome suggests that PSW successfully minimized selection bias by balancing the distribution of covariates across the classifications. Figure 1 illustrates the distributions of propensity scores before and after PSW. Before applying PSW, the mean propensity scores were significantly different between the fracture and non-fracture classifications (0.44 vs. 0.28, respectively; *p* < 0.001), indicating initial disparities in baseline variables between classifications. However, after PSW, the propensity score distributions became similar, with no statistically significant differences between the classifications (means of 0.29 ± 0.21 for the fracture classification and 0.34 ± 0.21 for the non-fracture classification; *p* = 0.499), confirming the effectiveness of PSW in balancing the groups at baseline (Figure 1).

### 3.2. Univariate and Multivariate Logistic Regression Analysis

To identify clinical factors associated with good postoperative gait ability, univariate and multivariate logistic regression analyses were conducted. The univariate logistic regression analysis revealed that gender (*p* < 0.001), age (*p* < 0.001), and muscle strength greater than grade 3 for hip flexion (*p* < 0.001) were significantly associated with good postoperative gait ability (Table 2).

The multivariate logistic regression analysis further identified four independent predictors of good postoperative gait ability. These included male gender (OR: 1.382, 95% CI: 1.225–1.560, *p* < 0.001), younger age (OR: 0.990 per year, 95% CI: 0.985–0.995, *p* < 0.001), preoperative independent ambulation with or without an assistive device (OR: 1.654, 95% CI: 1.093–2.502, *p* = 0.018 and OR: 1.531, 95% CI: 1.029–2.276, *p* = 0.037, respectively), and postoperative muscle strength greater than grade 3 for hip flexion (OR: 1.516, 95% CI: 1.177–1.953, *p* = 0.002) (Table 3).

## 4. Discussion

In this study, we identified several key clinical factors that were significantly associated with good postoperative walking ability after THA. Compared with previous studies, this study has several advantages. Our study leverages PSW to enhance the validity of findings by reducing selection bias and ensuring a well-balanced comparison between fracture and non-fracture classifications. This method has been increasingly utilized in observational studies to approximate randomized controlled trial conditions, thereby strengthening causal inference [22,23]. Prior studies examining predictors of gait recovery following THA have predominantly relied on traditional regression models, which may be prone to residual confounding [13,14]. By incorporating PSW, our study builds upon the existing literature and provides a more refined analysis of independent factors influencing postoperative mobility outcomes. This approach minimized selection bias by balancing covariates between the fracture and non-fracture classifications, thereby improving causal inference. However, while PSW reduces confounding and enhances comparability between groups, it does not fully replicate the conditions of a randomized controlled trial (RCT), as unmeasured confounders may still influence the observed associations [28,30,31]. This method is crucial due to the observational design and challenges in balancing patient classifications. The decision to categorize patients based on ‘fracture’ and ‘non-fracture’ status was guided by the need to account for different recovery trajectories and rehabilitation needs. Patients undergoing THA for fracture-related conditions typically have acute injuries and may have different postoperative outcomes than patients undergoing THA for non-fracture conditions such as osteoarthritis or avascular necrosis [20,21]. This categorization allowed for a more nuanced analysis of clinical predictors tailored to the specific characteristics of each condition. To further enrich the analysis, both univariate and multivariate logistic regression analyses were used: univariate analyses identified individual factors associated with good walking ability, while multivariate analyses adjusted for potential confounders to comprehensively identify independent predictors. This dual approach ensured that the identified predictors were not only statistically significant but also clinically meaningful, which is critical for translating research findings into real-world applications.

The classification of gait ability in this study, based on functional tasks such as stair climbing and home ambulation, aligns with clinical assessments of mobility post THA [18,32]. Although these measures primarily assess functional mobility, they are widely used as indicators of postoperative independence in rehabilitation settings. While biomechanical gait parameters such as stride length and joint kinematics were not incorporated in this study, they could provide additional insight into gait recovery. Nonetheless, functional recovery assessments remain critical for guiding rehabilitation planning and patient counseling. Our analysis identified four key clinical factors significantly associated with good postoperative gait ability: gender, age, preoperative gait ability, and postoperative muscle strength of hip flexion. Male patients are more likely to achieve good postoperative gait ability than females. This observation is consistent with previous studies suggesting that males generally have greater muscle mass and bone density, which may contribute to better functional recovery after THA [13,33,34]. Gender-specific considerations may be needed for postoperative rehabilitation, especially for female patients. Female patients generally exhibit lower baseline muscle mass and neuromuscular activation efficiency compared to males, which can lead to delayed functional recovery following THA. Implementing structured resistance training and neuromuscular re-education protocols tailored to female patients optimizes muscle recruitment patterns and improves gait mechanics, ultimately enhancing postoperative mobility outcomes [35,36]. Younger patients were more likely to demonstrate good postoperative gait ability. This finding aligns with the existing literature, which suggests that younger patients tend to exhibit enhanced musculoskeletal healing capacity, greater physiological reserve, and improved neuromuscular coordination, all of which facilitate early weight-bearing and contribute to improved functional mobility and self-reported gait ability following THA [13,14]. These advantages allow for faster rehabilitation progress and greater confidence in mobility, ultimately leading to superior recovery outcomes. This highlights the importance of considering age in the rehabilitation planning process, as older patients may require extended rehabilitation incorporating progressive resistance training, neuromuscular re-education, and structured balance exercises to optimize mobility and reduce fall risk. The ability to ambulate independently with or without an assistive device before surgery was strongly associated with good postoperative gait ability. This finding underscores the importance of preoperative functional status as a predictor of postoperative outcomes and supports the value of pre-rehabilitation programs aimed at improving functional capacity before surgery [13,37]. Enhancing preoperative gait ability through targeted exercises and training may help to improve postoperative recovery and overall functional outcomes. Patients with postoperative muscle strength greater than grade 3 for hip flexion of the operated limb demonstrated significantly better gait outcomes. This finding is consistent with previous studies that emphasize the importance of hip muscle strength for functional mobility and recovery after THA [38,39,40]. Weakness in hip flexor muscles post surgery can lead to compromised gait patterns and reduced mobility [41].

Leveraging the strengths of PSW, our study has identified these four key clinical factors associated with good postoperative gait ability following THA. Our identification of modifiable factors, particularly muscle strength, provides actionable insights for improving patient care. Among these, the importance of hip flexor muscle strength stands out as a modifiable factor that can be directly targeted through rehabilitation programs. These findings can guide clinicians in patient selection, preoperative counseling, and postoperative rehabilitation planning. Specifically, they underscore the potential benefits of implementing comprehensive rehabilitation programs focused on strengthening hip flexors in patients with THA.

### Limitations and Future Directions

Despite the strengths of our study, there are several limitations that should be acknowledged. First, the sample size, although sufficient for PSW analysis, is smaller than some larger-scale studies. However, PSW helped to mitigate this limitation by effectively balancing covariates and reducing bias, thereby enhancing the robustness of our findings. Second, our study focused on short-term gait ability following THA, assessing outcomes only up to two weeks post surgery. This limitation means that our findings may not fully capture long-term functional outcomes, as gait ability beyond the immediate postoperative period was not evaluated. Additionally, we did not include comprehensive gait analysis data due to the potential for pain-induced gait limitations and instability in the immediate postoperative period, which could result in inconsistent or misleading data [42]. Most previous studies that have conducted gait analysis after THA typically performed these assessments after at least one-month post surgery [43,44,45,46]. Our study focused on earlier postoperative outcomes, which precluded the use of such analyses. These factors led us to focus on more stable and consistently measurable outcomes in the early postoperative period. Acute postoperative pain and psychological factors, such as movement-related anxiety, may influence early gait recovery following THA, particularly in fracture patients. Higher pain levels can contribute to altered biomechanics and slower rehabilitation progress [47,48]. While pain and psychological status were not directly assessed in this study, future research should incorporate validated pain scales and psychological screening tools to better understand their impact on early mobility outcomes. Another limitation is the reliance on structured interviews and clinician-assessed gait ability rather than validated patient-reported outcome measures (PROMs) or standardized functional mobility tests. While clinician-assessed gait ability provides an objective measure of postoperative mobility, PROMs such as the EQ-5D or Oxford Hip Score could offer complementary insights into self-perceived recovery, pain levels, and activity limitations. Incorporating both clinician-based assessments and PROMs in future studies could provide a more comprehensive evaluation of functional recovery following THA, allowing for a multidimensional understanding of patient outcomes. Additionally, clinician-assessed gait ability, while providing an objective evaluation, may still be influenced by subjective interpretation and does not capture detailed biomechanical parameters. Future studies should integrate instrumented gait analysis or wearable sensor-based assessments to validate functional classifications against objective performance-based gait measures, ensuring a more precise evaluation of mobility outcomes.

To enhance accuracy in gait assessment, incorporating objective gait analysis tools or wearable sensor-based assessments may provide more precise and quantitative measures of postoperative recovery while minimizing subjective bias. These advanced techniques could offer valuable insights into kinetic and kinematic parameters of gait recovery and guide the development of targeted rehabilitation programs.

Future studies should consider multi-center, longitudinal studies with larger sample sizes to explore the long-term impact of targeted muscle-strengthening programs on gait recovery in THA patients. Additionally, biomechanical gait analyses in conjunction with larger sample sizes may help to clarify the role of hip abduction strength in gait recovery. Furthermore, prospective studies that incorporate PSW at the design stage could provide further insights into predictive factors and their implications for patient care, offering more tailored rehabilitation strategies to optimize outcomes.

## 5. Conclusions

Our study identified four key clinical predictors associated with good postoperative gait ability following THA: male gender, younger age, independent preoperative gait ability, and adequate postoperative hip flexor muscle strength. Among these, postoperative hip flexor strength was the most modifiable factor, highlighting the importance of targeted rehabilitation interventions. Strengthening programs focusing on hip flexor muscles in both preoperative and postoperative settings may enhance gait recovery and functional independence in THA patients. These findings provide clinically relevant insights into patient selection, rehabilitation planning, and functional prognosis following THA. Future research should explore personalized rehabilitation protocols based on these predictors and evaluate the long-term effectiveness of targeted strengthening programs in diverse patient populations.

## Figures and Tables

**Figure 1 jcm-14-01979-f001:**
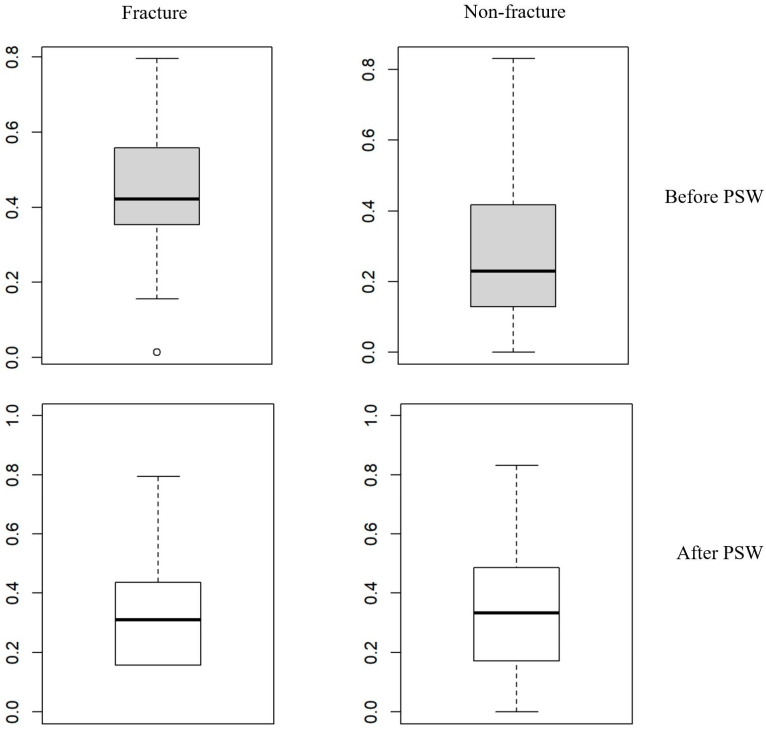
Distribution of propensity scores before and after propensity score weighting in fracture and non-fracture groups. This figure displays box plots illustrating the differences in variables between fracture and non-fracture groups in THA patients before and after propensity score weighting (PSW). The box-and-whisker plots show the first quartile (Q1 or 25th percentile) as the bottom line of the box, the median (Q2 or 50th percentile) as the middle line, and the third quartile (Q3 or 75th percentile) as the top line of the box. The whiskers extend to the minimum and maximum observations. Before PSW, the mean propensity scores were significantly different, with 0.44 ± 0.15 for the fracture group and 0.28 ± 0.18 for the non-fracture group (*p* < 0.001). After PSW, the mean propensity scores were not significantly different, with 0.29 ± 0.21 for the fracture group and 0.34 ± 0.21 for the non-fracture group (*p* = 0.499). This demonstrates the effectiveness of PSW in balancing the baseline characteristics between the two groups.

**Table 1 jcm-14-01979-t001:** Comparison of baseline characteristics between fracture and non-fracture groups in THA patients, before and after propensity score weighting.

	Before PSW				After PSW			
	Fracture	Non-Fracture	*p*-Value	SMD	Fracture	Non-Fracture	*p*-Value	SMD
*n*	73	148			84	150.5		
Gender = M (%)	32 (43.8)	77 (52.0)	0.252	0.165	50.9 (60.6)	72.9 (48.5)	0.356	0.246
Age (mean (SD))	68.33 (8.68)	61.73 (12.75)	<0.001 *	0.605	61.95 (10.94)	64.04 (12.26)	0.589	0.18
Preoperative gait ability (%)			0.029 *	0.518			0.334	0.472
Bed-ridden state	0 (0.0)	2 (1.4)			0.0 (0.0)	1.3 (0.9)		
Wheelchair mobile	1 (1.4)	5 (3.4)			2.0 (2.4)	4.2 (2.8)		
Assisted ambulation	5 (6.8)	5 (3.4)			4.3 (5.1)	9.6 (6.4)		
Independent ambulation with assistive device	4 (5.5)	29 (19.6)			28.3 (33.7)	22.1 (14.7)		
Independent ambulation without assistive device	63 (86.3)	107 (72.3)			49.5 (58.9)	113.3 (75.3)		
Duration of gait disturbance days (mean (SD))	52.67 (165.14)	114.74 (386.23)	0.19	0.209	205.40 (272.86)	97.46 (337.69)	0.32	0.352
Duration of rehabilitation days (mean (SD))	18.79 (7.71)	17.47 (13.60)	0.442	0.12	16.24 (8.40)	19.66 (20.21)	0.375	0.221
Interval between surgery to rehabilitation days (mean (SD))	6.03 (5.74)	6.73 (4.94)	0.348	0.131	7.45 (6.17)	6.72 (4.92)	0.522	0.131
Muscle strength of operated hip								
Flexion > grade 3 (%)	17 (23.3)	59 (39.9)	0.015 *	0.362	39.5 (47.0)	50.5 (33.5)	0.381	0.278
Abduction > grade 3 (%)	18 (24.7)	61 (41.2)	0.016 *	0.358	40.6 (48.3)	52.5 (34.9)	0.378	0.275
G group (%)	14 (19.2)	58 (39.2)	0.003	0.451	35.1 (41.8)	53.0 (35.2)	0.699	0.135

THA, total hip arthroplasty; PSW, propensity score weighting; SD, standardized difference; G group, good gait ability group. Continuous variables are presented as mean (standard deviation). Categorical variables are presented as *n* (%). *p*-values were calculated using independent *t*-tests for continuous variables and chi-square tests for categorical variables. * *p* < 0.05.

**Table 2 jcm-14-01979-t002:** Univariate analysis of factors associated with good gait ability after THA.

	G Group	P Group	*p*-Value
*n*	88.1	146.4	
Gender = M (%)	74.4 (84.4)	49.5 (33.8)	<0.001 *
Age (mean (SD))	57.03 (11.52)	67.06 (10.35)	<0.001 *
Preoperative ambulatory status (%)			0.171
Bed-ridden state	0 (0.0)	1.3 (0.9)	
Wheelchair mobile	0.7 (0.8)	5.5 (3.7)	
Assisted amulation	0.0 (0.0)	13.9 (9.5)	
Independent ambulation with assistive device	28.4 (32.2)	22.1 (15.1)	
Independent ambulation without assistive device	59.0 (67.0)	103.7 (70.8)	
Underlying disease (%)			0.451
Fracture	35.1 (39.8)	48.9 (33.4)	
Osteoarthritis	23.1 (26.3)	47.6 (32.5)	
Avascular necrosis	27.7 (31.5)	36.1 (24.6)	
Others	2.1 (2.4)	13.8 (9.4)	
Duration of gait disturbance days (mean (SD))	176.01 (259.08)	112.14 (349.72)	0.566
Duration of rehabilitation days (mean (SD))	14.07 (10.26)	21.06 (19.57)	0.050
Interval from surgery to rehabilitation days (mean (SD))	7.68 (4.76)	6.56 (5.73)	0.256
Muscle strength of operated hip			
Flexion > grade 3 (%)	56.0 (63.6)	34.0 (23.2)	<0.001 *
Abduction > grade 3 (%)	56.0 (63.6)	37.1 (25.3)	0.001 *

THA, total hip arthroplasty; G group, good gait ability group; P group, poor gait ability group; SD, standardized difference. Continuous variables are presented as mean (standard deviation). Categorical variables are presented as *n* (%). *p*-values were calculated using independent *t*-tests for continuous variables and chi-square tests for categorical variables. * *p* < 0.05.

**Table 3 jcm-14-01979-t003:** Multivariate analysis of factors associated with good gait ability after THA.

	Odds Ratio	CIL	CIH	*p*-Value
*n*				
Gender, male	1.382	1.225	1.560	<0.001 *
Age, year	0.990	0.985	0.995	<0.001 *
Preoperative gait ability				
Bed-ridden state	Ref			
Wheelchair mobile	1.189	0.752	1.881	0.459
Assisted ambulation	1.326	0.867	2.030	0.195
Independent ambulation with assistive device	1.654	1.093	2.502	0.018 *
Independent ambulation without assistive device	1.531	1.029	2.276	0.037 *
Underlying disease				
Fracture	Ref			
Osteoarthritis	1.078	0.946	1.228	0.262
Avascular necrosis	1.064	0.928	1.220	0.373
Others	0.889	0.735	1.075	0.225
Duration of gait disturbance.days	1.000	1.000	1.000	0.916
Duration of rehabilitation.days	0.999	0.995	1.002	0.475
Interval from surgery to rehabilitation.days	1.003	0.989	1.017	0.665
Muscle strength of operated limb				
Flexion > grade 3 vs. ≤grade 3	1.516	1.177	1.953	0.002 *
Abduction > grade 3 vs. ≤grade 3	0.791	0.622	1.005	0.057

THA, total hip arthroplasty; CIL, confidence interval lower bound; CIH, confidence interval higher bound. The odds ratios were calculated using multivariable logistic regression analysis with G group (good gait ability) as the dependent variable, adjusting for all variables shown in the table. Ref, reference category. * *p* < 0.05.

## Data Availability

The data presented in this study are available from the corresponding authors upon reasonable request.

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
