# Peer review of "Predictive Factors for Gait Recovery in Patients Undergoing Total Hip Arthroplasty: A Propensity Score Weighting Analysis"

_jcm, 2025, doi:10.3390/jcm14061979_

Round 1
Reviewer 1 Report
Comments and Suggestions for Authors
The current study implemented an analysis (propensity score weighting) to improve the robustness of statistical analysis to identify modifiable factors when comparing the effect of total hip arthroplasty across fracture and non-fracture patient conditions on gait ability. Although I commend the authors for applying the PSW analysis to improve the robustness of their outcome measures, the value of the current study in terms of gait performance and recovery is limited. Thank you for the opportunity to review your manuscript. Below are my main comments.
Introduction
- The specific knowledge gap that the current study seeks to address is unclear. The authors mention many studies attempt to identify predictors of recovery post arthroplasty but do not mention the current state of literature. Please outline how current key factors are modified.
- The introduction is limited and vague and should include a more in-depth appraisal of the gait literature in the context of their research question.
- What exactly are your ‘modifiable’ factors of interest in the current study and what is your hypothesis? These should be explicitly stated.
- Have the authors considered cluster analysis methods in their literature review?
- The authors classify participants into disease conditions and fracture conditions, but the introduction does not distinguish between these two groups suggesting similar THA outcomes. This should be elaborated on as one would expect large differences between these conditions on recovery trajectories (as nicely justified in their method section) but not explained explicitly on gait recovery. One large difference would be participation and sustained engagement in rehabilitation interventions. However, how do these differences in THA-patient conditions differ during gait performance?
Methods
- Please include a sample size estimation.
- How does physical activity influence recovery trajectory on gait performance? Did you consider the PASE questionnaire or any other markers of physical activity?
- Major limitation: assessment of gait performance is severely limited in justification and no reference is provided on the method used (e.g., structured interview). Why weren’t patient-reported questionnaires used? Moreover, patients’ perceived mobility is not representative of their actual mobility based on in-clinic and free-living performance. I’m surprised that only two ‘gait ability’ categories were selected and identified.
- How long were the structured interviews?
- Gait metrics such as stride length and range of motion are mentioned in the introduction, but no mention of self-reported gait ability and the utility of patient-reported gait ability is explained. In addition, mobility estimates should be included in the introduction, not just gait. Mobility should also be defined for the reader.
- I strongly recommend making a table, or a section that clearly identified the clinical predictors of interest and a rationale of selecting these predictors as it is unclear why they are selected.
- What rehabilitation was performed after arthroplasty?
Results
- One cannot have a fractional (0.5) patient - line 190 and throughout the results section.
- Abduction strength is trending towards significance (Table 3; p=0.057) and the odds ratio is in the other direction compared to flexion strength. This is unexpected since abduction strength should contribute to gait ability/mobility. This warrants discussion.
Discussion
- Lines 256-258 require a reference.
- I appreciate the definitions provided in regard to good and bad gait ability. However, these belong in the methods section. More importantly, is stair ascending and descending as well as ambulating at home the only measures of “ability to perform essential activities”? One would argue these are more mobility instead of gait measures. Lines 271-275 must include references as your gait ability measure is only estimated using your definitions (which also don‘t have references). These lines are also very vague to the reader.
- Have you considered other self-reported questionnaires and clinical tests of mobility/gait performance that the literature commonly uses (EQ5D, oxford knee score, four-meter-walk test, timed up and go test). I recommend using the term ‘self-perceived’ gait ability going forward when referring to gait performance.
- How does pain contribute to your outcome measures between the two groups. What was the pain level pre- and post-THA, (how about surgical expectations, depression). This is critical considering the acute and chronic aspects of the fracture and non-fracture conditions, respectively.
- Please elaborate on what gender-specific considerations may be for post-operative rehabilitation for female patients as commented on line 281-282.
- Lines 283-286 need references and elaboration. This vague statement is also not directly related to gait performance. For example, how do faster healing rates contribute to better self-reported gait ability?
- It would be helpful for the authors to define mobility and gait ability.
- Could it not be, since gait ability was determined using patient-reported data via interviews, that one’s perception of performance may be over or under-estimated? How well do you expect your findings to be associated with performance-based gait measures? Please discuss.
- For, lines 286-288, what would more ‘intensive’ or prolonged rehabilitation look like? The challenge here is adherence to rehabilitation participation which is particularly poor among older adults and those with greater mobility limitations pre-operatively.
- Please describe the strength grading in more detail and the rationale for grading 3 as a threshold of importance for the reader in the methods section. This is especially important as the majority of your discussion focuses on hip flexion strength.
- The discussion section requires more work in terms of how the results and the PSW analysis fits into the body of literature.
- Limited discussion on the findings of sex differences.
- I suggest softening the claim that the PSW can ‘simulate’ the conditions of a randomized control trial. There are aspects the PSW mimics, but the authors should mention what the PSW lacks compared to a randomized control trial (e.g., unmeasured confounders) as they bring this comparison up multiple times.
Reviewer 2 Report
Comments and Suggestions for Authors
Dear Authors,
Special thanks for your manuscript submission and to the editors for providing the opportunity to review this work.
General comment
The study, titled Predictive Factors for Gait Recovery in Patients Undergoing Total Hip Arthroplasty: A Propensity Score Weighting Analysis,' aims to identify clinical predictors, especially modifiable ones, associated with postoperative gait recovery in total hip arthroplasty (THA) patients, utilizing propensity score weighting (PSW) to control confounding factors. The study recruited 221 patients included in this study. Despite the relevance of the topic, the paper has several comments (indicated below) that need to be addressed before it can be considered for publication.
Specific comments
- This research has the potential to interest the JCM readership but requires significant improvement.
- Lines 68-74: The aim of the study needs to be rewritten for more accuracy.
- Lines 143-174: It is preferable to move to the description of experiments subsection.
- Page 5 is empty; please revise it.
- In future work, the pre- and postoperative gait ability should be assessed using an accurate tool or by biomechanical gait analysis for more accuracy.
- Lines 308-322: It is preferable to introduce a separate section for limitations and future work before the conclusions section.
- While the paper is well-illustrated and presented, it should be rewriting the conclusions to focus on the most relevant results to the study's aim.
Overall,
- The experimental protocol and procedures are clearly described and well-structured.
- The cited references are appropriate and relevant to the research topic.
- This scientific work presents interesting and valuable insights that will be beneficial to readers of the journal.
